# Long Non-Coding RNAs in Colorectal Cancer: Navigating the Intersections of Immunity, Intercellular Communication, and Therapeutic Potential

**DOI:** 10.3390/biomedicines11092411

**Published:** 2023-08-28

**Authors:** Nikolay K. Shakhpazyan, Liudmila M. Mikhaleva, Arcady L. Bedzhanyan, Nikolay K. Sadykhov, Konstantin Y. Midiber, Alexandra K. Konyukova, Andrey S. Kontorschikov, Ksenia S. Maslenkina, Alexander N. Orekhov

**Affiliations:** 1Avtsyn Research Institute of Human Morphology, Petrovsky National Research Center of Surgery, 119435 Moscow, Russia; mikhalevalm@yandex.ru (L.M.M.); drawnman@mail.ru (N.K.S.); midiberkonst@gmail.com (K.Y.M.); have.to.study@yandex.ru (A.K.K.); andreistr.ru@mail.ru (A.S.K.); ksusha-voi@yandex.ru (K.S.M.); alexandernikolaevichorekhov@gmail.com (A.N.O.); 2Department of Abdominal Surgery and Oncology II (Coloproctology and Uro-Gynecology), Petrovsky National Research Center of Surgery, 119435 Moscow, Russia; arkady.bedzhanyan@gmail.com; 3Laboratory of Angiopathology, Institute of General Pathology and Pathophysiology, 125315 Moscow, Russia; 4Institute for Atherosclerosis Research, 121096 Moscow, Russia

**Keywords:** colorectal cancers (CRCs), immune cells, tumor environment (TME), long non-coding RNAs (lncRNAs)

## Abstract

This comprehensive review elucidates the intricate roles of long non-coding RNAs (lncRNAs) within the colorectal cancer (CRC) microenvironment, intersecting the domains of immunity, intercellular communication, and therapeutic potential. lncRNAs, which are significantly involved in the pathogenesis of CRC, immune evasion, and the treatment response to CRC, have crucial implications in inflammation and serve as promising candidates for novel therapeutic strategies and biomarkers. This review scrutinizes the interaction of lncRNAs with the Consensus Molecular Subtypes (CMSs) of CRC, their complex interplay with the tumor stroma affecting immunity and inflammation, and their conveyance via extracellular vesicles, particularly exosomes. Furthermore, we delve into the intricate relationship between lncRNAs and other non-coding RNAs, including microRNAs and circular RNAs, in mediating cell-to-cell communication within the CRC microenvironment. Lastly, we propose potential strategies to manipulate lncRNAs to enhance anti-tumor immunity, thereby underlining the significance of lncRNAs in devising innovative therapeutic interventions in CRC.

## 1. Introduction

Colorectal cancer (CRC), encompassing cancers of the colon and rectum, is one of the most prevalent malignancies worldwide and a leading cause of cancer-related death [1]. This complex disease, arising from a blend of genetic and environmental factors, manifests as a substantial challenge in oncology due to late-stage diagnosis and resistance to conventional therapies. Understanding the intricate web of CRC requires a deep exploration of its underlying mechanisms, including the significant role of inflammation.

Inflammation is pivotal in CRC pathogenesis, from the inception of aberrant crypt foci to advanced metastatic disease [2,3,4,5]. It not only fuels the initiation and progression of CRC but also fundamentally shapes the tumor microenvironment and the immune response to the tumor [6]. This intertwining of inflammation and cancer has unveiled new frontiers in treatment, including the potential role of long non-coding RNAs (lncRNAs).

Recent evidence has revealed lncRNAs, non-coding RNAs exceeding 200 nucleotides in length, as key players in various biological processes, including inflammation and cancer [7]. Dysregulated lncRNAs are increasingly being recognized as significant contributors to the pathogenesis of CRC immune evasion, and the response to therapy. Moreover, lncRNAs appear to be intricately connected to the inflammatory processes driving CRC, marking them as potential predictive biomarkers for treatment response [8].

The interest in lncRNAs extends to their potential role in enhancing the efficacy of existing treatments, such as immune checkpoint blockade therapies like anti-PD-1/PD-L1 [9,10]. While these therapies have shown promise, their effectiveness varies between individuals, and many patients remain unresponsive. The exploration of lncRNAs in this context might unveil novel avenues for identifying the patients who are most likely to benefit from these immunotherapies and open doors for innovative therapeutic targets [11].

In this review, we will conduct a comprehensive examination of the multifaceted roles of lncRNAs in colorectal cancer, with a strong focus on their involvement in modulating immune responses and mediating intercellular communication within the tumor microenvironment. We will delve into their impact on the response to immunotherapy, the distinctive role of inflammation in the pathogenesis of CRC, its interplay with lncRNA function, and the potential strategies for manipulating lncRNAs to enhance anti-tumor immunity. Furthermore, we will explore the prospects of the use of lncRNAs as diagnostic and prognostic biomarkers in colorectal cancer.

## 2. The Fundamentals of Long Non-Coding RNAs

Long non-coding RNAs (lncRNAs) are RNA molecules that exceed 200 nucleotides in length and are characterized by their lack of classical protein-coding capacity. These critical biomolecules have recently been brought to the forefront of molecular biology due to their essential roles in regulating several biological processes such as gene expression, cellular differentiation, and development [11].

To further elucidate the intriguing nature of lncRNAs, it is vital to understand their genomic classification, which can provide insight into their nature and, to some extent, their function. As shown in Table 1, lncRNAs can be classified according to the genomic locations of their genes.

This classification is valuable for understanding potential regulatory roles and interactions within cellular processes. While some lncRNAs may function in close conjunction with neighboring protein-coding genes, others might exert influence over more distant genomic regions or engage in complex interactions that are not solely dictated by their location [12]. Further research is needed to fully grasp the diverse functions of lncRNAs, considering factors such as cell type, developmental stage, and post-transcriptional modifications.

Expanding upon their roles, lncRNAs also play significant roles in various cellular processes, including gene expression regulation, RNA interactions, RNA–protein interactions, structural roles, and signaling regulation. An overview of these functions and mechanisms is presented in Table 2.

The multifaceted nature of lncRNAs and the context-dependent specificity of their functions may lead to real-world scenarios in which the complexities extend beyond this categorization [18,19,20,21,22,23,24,25,26,27,28,29].

Shifting our focus to the broader context of oncogenesis, lncRNAs modulate key cellular processes like proliferation, apoptosis, migration, and invasion. They can potentially instigate the transformation of normal cells into malignant ones [30,31,32]. Moreover, lncRNAs significantly shape the anti-tumor immune landscape by interacting with DNA, RNA, and proteins and by modulating the expression of immune response genes [33].

The mechanisms through which lncRNAs function are as diverse as their roles. For instance, lncRNAs like HOTAIR can promote or hinder the assembly of transcriptional machinery at gene promoters, interacting with the Polycomb Repressive Complex 2 (PRC2) and thereby silencing target genes [34]. Certain lncRNAs also influence post-transcriptional processes, such as mRNA splicing, stability, and translation with MALAT1 [35,36,37]. Furthermore, lncRNAs can function as molecular scaffolds or as molecular decoys, like the lncRNA GAS5 does [38,39].

Delving into their role in cellular communication, lncRNAs play a critical role within the tumor microenvironment. They can be encapsulated into extracellular vesicles such as exosomes and secreted from cells, altering various cellular functions inside recipient cells. This intercellular lncRNA exchange has been associated with aspects of tumor biology like immune modulation, angiogenesis, metastasis, and therapy resistance [40].

In conclusion, this chapter provides general information about this versatile and complex class of biomolecules. The insights laid out will serve as a foundation for the forthcoming discussion on the role of lncRNAs in processes associated with oncogenesis in colorectal cancer. The text is intended to offer a comprehensive yet concise overview, setting the stage for a deeper exploration of the topic.

It would be valuable to mention the role of the sources of information when investigating issues connected with lncRNAs. This review is rooted in a synthesis of data from the literature, supplemented by various lncRNA databases, illustrating a synergistic relationship between the sources in the literature and databases in the context of bioinformatics. Various lncRNA–disease association data resources have played an instrumental role in capturing associations with diseases, including CRC. LncRNADisease, now upgraded to LncRNADisease v2.0, includes associations with 529 diseases [41]. Lnc2Cancer is a manually curated database focusing specifically on lncRNA–cancer associations [42], while MNDR covers a wide array of ncRNA–disease associations in mammals [43]. Additionally, resources like LNCipedia [44], lncRNAWiki [45], and lncRNome [46] offer sequence and annotation information, complementing our understanding of the attributes of lncRNAs. Comprehensive platforms such as LncTarD 2.0 have further enhanced the capacity to discern the properties of cancer stem cells and potential clinical applications [47]. Community collaboration is a salient feature, exemplified by databases like lncRNAWiki, which integrate data from various sources [45]. In our review, the sources in the literature often mention these databases, facilitating the meticulous selection of lncRNAs. Tools like RNA-seq/microarray and single-cell web tools, provided by platforms such as LncTarD 2.0, enable the customizable analysis of lncRNA–target regulations in diseases [47]. In conclusion, these databases act as useful platforms in navigating the intricate landscape of lncRNAs and are tailored for general use, cancer research, and other specialized domains [41,42,43,44,45,46,47]. They offer intuitive interfaces, integrated literature searches, predictive tools, and experimental data, making them indispensable for researchers. The integration of such analyses substantiates the selected molecules’ significance, contributing to an overarching understanding of their roles in CRC. Future enhancements could further foster a comprehensive and accessible repository, enhancing our understanding of lncRNAs’ roles in human diseases, including CRC [41,42,43,44,45,46,47].

## 3. Interplay of lncRNAs and Consensus Molecular Subtypes (CMSs) in CRC

The Consensus Molecular Subtypes (CMSs) of CRC offer an invaluable framework for delving into the intricate molecular landscape of this heterogeneous disease [48,49,50]. The distinct biological profiles of CMS1 (MSI immune), CMS2 (canonical), CMS3 (metabolic), and CMS4 (mesenchymal) each carry implications for diagnosis, prognosis, and therapeutic intervention (Table 3).

The CMS1 (MSI immune) subtype, characterized by high microsatellite instability (MSI-H), showcases a hypermutated phenotype that often incites robust immune responses. Within this subtype, both lncRNAs, HOTAIR and LINK-A, are involved in oncogenic activities, focusing on immune activation and hypermutation. HOTAIR, an antisense lncRNA, regulates gene expression by interacting with the Polycomb Repressive Complex 2 (PRC2) and LSD1, modulating H3K27 methylation and thereby affecting gene silencing. This has consequences for PTEN methylation and pathways like PI3K/p-AKT/p-MDM2/p53, and PI3K/AKT/mTOR in tumorigenesis [51,52,53,54]. LINK-A, an intergenic lncRNA, modulates pathways, attenuating PKA activity on TRIM71 and causing AKT hyperactivation, leading to tumorigenesis [55,56,57,58] (Table 4).

The CMS2 (canonical) subtype is marked by chromosomal instability and WNT and MYC signaling and is associated with the lncRNAs CCAT1 and CRNDE. These lncRNAs in this subtype are engaged in oncogenic mechanisms, such as miRNA sponging and pathway modulation. CCAT1, an intergenic lncRNA, might interact with EZH2 to mediate chromosome looping with CTCF, affecting the c-Myc promoter and leading to gene silencing [59,60]. CRNDE, another intergenic lncRNA, is involved in the Wnt/β-catenin and other pathways, promoting cell growth [61,62,63] (Table 4).

The CMS3 (metabolic) subtype, distinguished by metabolic dysregulation and prevalent KRAS mutations, includes lncRNAs like lncRNA-ATB and RP11-462C24.1. Both lncRNAs in this subtype are intergenic and contribute to metabolic reprogramming, either through oncogenic or oncosuppressive functions. lncRNA-ATB may induce the epithelial-to-mesenchymal transition (EMT) by sponging miR-200 family members, while RP11-462C24.1 is associated with oxidative phosphorylation regulation and the upregulation of HSP70 [64,65,66,67,68,69] (Table 4).

In the CMS4 (mesenchymal) subtype, which is known for TGF-β activation, stromal invasion, and angiogenesis, the lncRNAs H19 and lincRNA-p21 are vital. These lncRNAs are linked with EMT, angiogenesis, inflammation, and matrix remodeling. H19 can upregulate HMGA2 by sponging the let-7 microRNA, and lincRNA-p21 can induce EMT in response to TGF-β, augmenting the invasive and metastatic potential of cancer cells [70,71,72,73,74,75,76,78] (Table 4).

Interestingly, these lncRNAs not only play a variety of roles in cellular processes but also demonstrate subtype specificity in CRC. For instance, HOTAIR’s association with CMS1 and its potential role in immune evasion, or CCAT1′s correlation with CMS2 and its influence on cell proliferation and invasion, underline the versatility and specificity of lncRNAs. As we continue to learn more about their roles and functions, lncRNAs are emerging as potential therapeutic targets. Their distinct expression in different CMS subtypes could be exploited to develop personalized therapeutic strategies, such as inhibiting overexpressed lncRNAs like HOTAIR in CMS1 or CCAT1 in CMS2. Furthermore, lncRNAs like H19 in CMS4 have been associated with a poor prognosis, suggesting their potential utility as diagnostic or prognostic biomarkers [6].

Another fascinating aspect of lncRNAs is their influence on drug resistance in CRC, making them a crucial focus for improving treatment outcomes. Additionally, lncRNAs have been found to interact with microRNAs, influencing their function and contributing to complex regulatory networks. For instance, the interaction of H19 with the let-7 miRNA in CMS4 deregulates its targets, promoting the epithelial-to-mesenchymal transition and stemness [79]. As the field of lncRNA research continues to develop, it offers promising new possibilities for understanding the molecular intricacies of CRC subtypes and refining therapeutic strategies to enhance patient outcomes.

## 4. The Role of Immunity and Inflammation in CRC Tumor Stroma

LncRNAs have emerged as pivotal regulatory elements across a broad spectrum of biological processes. Their roles are especially conspicuous within the intricate orchestration of the tumor microenvironment (TME), a crucial aspect of cancer’s multifaceted architecture. In CRC, lncRNAs are known to notably modulate fundamental cellular mechanisms and reciprocal interactions, thereby shaping the immune landscape of the TME. Consequently, elucidating the influence of lncRNA-guided processes on inflammation, immune responses, and metabolic reprogramming within the TME is crucial. This section sheds light on the main orchestrators involved in shaping the immunoregulatory milieu surrounding tumor cells (Figure 1).

Inflammation plays a pivotal role in the TME of CRC. Cancer-associated fibroblasts (CAFs) and tumor-associated macrophages (TAMs) significantly contribute to this process, secreting pro-inflammatory cytokines such as interleukin-6 (IL-6) and tumor necrosis factor-alpha (TNF-α). Paradoxically, these cytokines can foster chronic inflammation, potentially promoting tumor progression [80]. Meanwhile, the ensuing inflammation can stimulate the recruitment and activation of immune cells capable of eliminating tumor cells.

The tumor stroma in CRC fosters the establishment of an immunosuppressive environment through various mechanisms. Notably, stromal cells, chiefly CAFs, secrete immunosuppressive factors such as transforming growth factor-beta (TGF-β) and interleukin-10 (IL-10), which inhibit T cell activity while enhancing regulatory T cell (Treg) functions [81,82]. Similarly, TAMs secrete factors like TGF-β, IL-10, and programmed cell death ligand-1 (PD-L1), collectively inhibiting the function of cytotoxic T cells and promoting the activity of Tregs [83,84].

Moreover, chronic exposure to tumor antigens and inflammatory signals in the TME can drive a state of T cell exhaustion which is characterized by diminished effector functions and the persistent expression of inhibitory receptors, including programmed cell death protein-1 (PD-1) and cytotoxic T-lymphocyte-associated protein 4 (CTLA-4). This state renders T cells less effective in eliminating cancer cells, thus facilitating immune evasion [85].

Another significant immunosuppressive mechanism in the TME involves the recruitment of regulatory immune cells. The tumor stroma can attract immunosuppressive cell types such as myeloid-derived suppressor cells (MDSCs) and Tregs, which impede the activity of cytotoxic T cells and natural killer (NK) cells, further facilitating tumor immune evasion [86,87].

Metabolic reprogramming is a defining feature of the TME. Both tumor cells and stromal cells can reshape the metabolic landscape of the TME, leading to conditions like hypoxia and nutrient deprivation. These conditions can adversely affect immune cell function and survival. For instance, tumor cells and CAFs can deplete vital nutrients like glucose and amino acids, thereby inhibiting T cell function [86].

CAFs, as primary stromal cells, play a crucial role in remodeling the extracellular matrix (ECM). This remodeling can construct a physical barrier that hinders immune cell infiltration and access to tumor cells. Therefore, understanding these mechanisms and interactions is vital for devising therapeutic strategies to counteract the immunosuppressive TME and boost antitumor immunity in CRC.

Within CRC, the TME can induce various forms of cell polarization to enhance the immunosuppressive state, thereby aiding the tumor in immune evasion. Macrophages within the TME often adopt an M2 polarization state known as “alternatively activated”. These M2 TAMs foster tissue repair, angiogenesis, and immune suppression by producing anti-inflammatory cytokines such as IL-10 and TGF-β and expressing high levels of immune checkpoint molecules like PD-L1, thereby inhibiting T cell function [87].

Conventional CD4+ T cells within the TME can be polarized into Tregs, identified via the expression of the transcription factor FOXP3 [83]. Tregs suppress the immune response by inhibiting the function of cytotoxic CD8+ T cells and other immune cells. Additionally, chronic exposure to antigens within the TME can instigate T cell exhaustion, which is characterized by upregulated inhibitory receptors like PD-1 and CTLA-4 and diminished effector functions [88].

MDSCs, a heterogeneous group of immature myeloid cells, can suppress the function of T cells through multiple mechanisms, including the production of immunosuppressive factors like arginase-1, nitric oxide, and reactive oxygen species [89,90]. Tumor-induced factors can also render dendritic cells (DCs) tolerogenic, diminishing their ability to activate T cells and potentially inducing Tregs or anergic T cells. These cellular polarization processes assist the colorectal tumor in establishing an immunosuppressive environment, thus contributing to immune evasion [91]. However, these processes also provide potential targets for therapeutic intervention, such as strategies aimed at reprogramming TAMs toward an M1 (pro-inflammatory and tumoricidal) phenotype, inhibiting Tregs, or reversing T cell exhaustion.

A summary of the immune interactions within the tumor stroma can be found in Table 5.

## 5. The Role of LncRNAs in Tumor–Stroma Immune Interplay via Extracellular Vesicles and Exosomes

The complexity and heterogeneity of tumors originate from the intricate network of the tumor microenvironment, encompassing not only malignant cells but also stromal components such as fibroblasts, immune cells, and endothelial cells, together with extracellular elements. Recent advancements in the field of cancer biology spotlight the pivotal role of lncRNAs in regulating a myriad of cellular processes within the tumor microenvironment. Particularly, lncRNAs packed into extracellular vesicles (EVs), including exosomes, have been identified as significant players in orchestrating the nuanced interplay within the tumor microenvironment [92,93].

Extracellular vesicles, especially exosomes, are diminutive membranous vesicles with diameters ranging from 30 to 150 nm. A variety of cell types secrete these vesicles, which act as essential mediators of intercellular communication by transporting and transferring a diverse array of biological molecules, including proteins, lipids, and different RNA types such as lncRNAs, to recipient cells. The selective incorporation of lncRNAs into exosomes is a process that remains to be fully decoded. Nevertheless, existing studies have proposed that specific sequence motifs, RNA-binding protein interactions, and distinct lncRNA modifications might guide their preferential packaging into exosomes [94,95]. Such vesicular lncRNAs are shielded from extracellular RNase activity, ensuring their stable transfer and subsequent uptake by recipient cells.

Within the tumor microenvironment, cancer cells release exosomes laden with lncRNAs, which can be internalized by stromal and immune cells. The lncRNAs delivered to these recipient cells can influence their functions, consequently molding a tumor-supportive microenvironment [96]. For instance, lncRNAs can manipulate immune cell functionality, induce pro-inflammatory cytokine secretion, and trigger the recruitment of immunosuppressive cells [96,97,98], thus fostering conditions conducive to tumor progression. Several lncRNAs detected within extracellular vesicles such as microvesicles and exosomes have been discovered to affect inflammation and immune regulation in cancer. Some notable lncRNAs are detailed below.

CCAT1 (Colon-Cancer-Associated Transcript 1): CCAT1, a long non-coding RNA found in exosomes, is implicated in various types of cancer, including CRC. CCAT1 impacts inflammation, angiogenesis, and immune regulation within the tumor microenvironment (TME), promoting immune cell polarization and the release of pro-inflammatory cytokines. CCAT1′s influence extends to pancreatic cancer, where it mediates angiogenesis through the microRNA-138-5p–high mobility group A1 (miR-138-5p/HMGA1) axis in exosomes derived from cancer cells. CCAT1 may similarly impact the TME in CRC, influencing immune interactions and disease progression [99]. Studies have also associated the upregulation of CCAT1 with cigarette-smoke-induced airway inflammation, suggesting its involvement in inflammatory responses in diverse cell types, such as intestinal epithelial cells. Thus, CCAT1 appears to play a critical role in the immune and inflammatory responses of CRC, potentially through its release in extracellular vesicles like exosomes, underscoring the need for further research [100].

CCAT2 (Colon-Cancer-Associated Transcript 2): CCAT2 is an exosomal long non-coding RNA primarily connected to CRC, and it facilitates tumor progression via PI3K/AKT/mTOR signaling. In addition to bolstering cancer cell proliferation, migration, and invasion in the context of CRC, CCAT2′s upregulation in CRC tissues enhances cell growth and metastasis by interacting with TAF15 to stimulate RAB14 transcription. This interaction subsequently triggers the AKT/GSK3β signaling pathway, impacting the cell cycle, migration, and apoptosis. Furthermore, insights into other cancer types are provided through examples like tamoxifen-resistant MCF7 cells in which CCAT2 modulates the hsa-miR-145-5p/AKT3/mTOR axis, controlling cell behavior. This observation reflects common phenomena in cancer, such as signaling pathways, which are present across different cancer types. Thus, CCAT2 plays a pivotal role in the tumor–stroma immune interplay within the tumor microenvironment, holding potential as a diagnostic and prognostic biomarker for CRC treatment while also offering insights into other cancer contexts [101,102].

CRNDE (Colorectal Neoplasia Differentially Expressed): Another long non-coding RNA that is upregulated in exosomes from CRC cells. It influences inflammation and immune evasion by stimulating the release of immunosuppressive factors, inducing T cell exhaustion, and recruiting regulatory immune cells. CRNDE intensifies inflammation by activating NF-κB and JAK/STAT signaling pathways involved in the interplay between tumor and stromal immune cells. The exosome-transmitted CRNDE-h isoform participates in T helper 17 (Th17) cell differentiation, which correlates with CRC progression. CRNDE-h interacts with the PPXY motif of RORγt, a transcription factor, thereby inhibiting the Itch-mediated ubiquitination and degradation of RORγt. By mediating the tumor–stroma immune interplay via exosomes, CRNDE may represent a potential target for CRC immunotherapy [61,103,104].

H19: A long non-coding RNA that plays a noteworthy role in the tumor–stroma immune interplay in CRC, being modulated via extracellular vesicles/exosomes. Emanating from diverse cancer cells, H19 is implicated in inflammation, stimulating pro-inflammatory cytokine release and extracellular matrix remodeling. Studies indicate that tumor necrosis factor-α (TNF-α) upregulates H19, which then heightens inflammatory cytokine levels. In CRC, H19 may facilitate interactions with immune cells within the tumor stroma, thereby influencing disease progression. Moreover, H19 is involved in the STAT3 pathway, which is critical for cancer aggressiveness, potentially impacting the tumor microenvironment. Therefore, the examination of H19 in the context of CRC could yield valuable insights into tumorigenesis, inflammation, and potential therapeutic strategies [105,106,107].

HOTAIR (HOX Transcript Antisense RNA): Found in exosomes from various cancer cells, HOTAIR is an oncogenic non-coding RNA linked to tumor grade and prognosis in several carcinomas, notably breast cancer. In CRC (CRC), HOTAIR-bearing exosomes derived from tumors influence B cells toward a regulatory role characterized by programmed cell death-ligand 1 (PDL1), thereby suppressing CD8+ T cell activity. This underlines the possible role of exosomal HOTAIR in stimulating immune cell recruitment and mediating tumor–stroma immune interactions. Further, HOTAIR may influence the release of pro-inflammatory cytokines and spur extracellular matrix remodeling, aiding in the progression of CRC. Consequently, HOTAIR presents as a prospective target for enhancing immune responses in treatment-resistant CRC patients [108,109,110].

HULC (Highly Upregulated in Liver Cancer): Overexpressed in exosomes from CRC and liver cancer, HULC accelerates the progression of CRC by targeting miR-613 to stimulate cell proliferation and inhibit apoptosis. It may also impact the immune response by binding to miR-488, thereby enhancing EZH2-H3K27me3 enrichment at the p53 promoter region, suppressing p53 expression, and promoting tumor growth and metastasis. Given its integral role in cancer cell survival and immune interactions, HULC could potentially serve as an effective therapeutic target for CRC [111,112].

LINC00461: Found in exosomes from lung cancer and multiple myeloma cells, LINC00461 contributes to creating an immunosuppressive tumor microenvironment. LINC00461′s expression in CRC is inconsistent; some studies show that its upregulation promotes tumor growth and proliferation via the miR-323b-3p/NFIB axis, while others suggest that it downregulates tumor progression via miR-141/PHLPP2. LINC00461′s complex interactions with miRNAs modulate various cellular processes, including migration, invasion, and the epithelial–mesenchymal transition, thereby influencing CRC development and immunity. Additionally, LINC00461 functions as a competitive endogenous RNA (ceRNA) for PHLPP2, a colon cancer tumor suppressor. Hence, understanding LINC00461 dynamics could provide insights into CRC progression and immune regulation [113,114].

lnc-ATB (lncRNA Activated By TGF-β): Identified in exosomes from breast cancer cells, lnc-ATB stimulates the release of pro-inflammatory cytokines and recruits immune cells. In CRC, upregulated lnc-ATB enhances metastasis, represses E-cadherin, and triggers the epithelial–mesenchymal transition (EMT). It targets pathways involving CDK2 and miR-200c, promoting cell proliferation and reducing apoptosis. Additionally, lnc-ATB suppresses miR-141-3p, further fostering the progression of cancer. The increased expression of lnc-ATB correlates with a larger tumor size, vascular invasion, and lymph node metastasis in CRC, and its regulation of β-catenin maintains CRC cell stemness. Elevated plasma lnc-ATB has been proposed as a significant biomarker for distinguishing CRC patients, emphasizing its potential role in CRC diagnosis, prognosis, and immune interplay [115,116,117].

lnc-EGFR: lnc-EGFR, which is present in exosomes derived from CRC (CRC) and other carcinomas, could potentially facilitate immune evasion via the EGFR signaling pathway, possibly inducing T cell exhaustion. Observational studies have implicated lnc-EGFR in encouraging Treg differentiation and thwarting cytotoxic T lymphocyte-mediated killing, thereby augmenting tumor immune escape. This lncRNA, which has been observed to be upregulated in Tregs infiltrating hepatocellular carcinoma, appears to foster Treg differentiation via binding to the EGFR receptor and enhancing downstream signaling. A correlation has been found between high lnc-EGFR expression and increased tumor size, which suggests that lnc-EGFR may play a crucial role in Treg differentiation and the suppression of antitumor T cell responses in CRC patients. Thus, lnc-EGFR could potentially provide valuable insights into the tumor–stroma immune interaction via extracellular vesicles in CRC [118,119].

MALAT1 (Metastasis-Associated Lung Adenocarcinoma Transcript 1): Identified in exosomes from various cancer cells, MALAT1 is associated with modulating T cell function and inducing the release of pro-inflammatory cytokines, thus contributing to metabolic reprogramming and immune cell polarization. Research on triple-negative breast cancer shows a correlation between MALAT1 upregulation, tumor size, and lymph node metastasis. It is hypothesized that MALAT1 could inhibit both innate and adaptive immune responses by targeting specific microRNA pathways. Given its potential role in immune evasion and inflammation, a deeper understanding of MALAT1 function could elucidate the intricacies of tumor–stroma immune interaction via exosomes in CRC, possibly paving a way for novel therapeutic strategies that target the NF-κB signaling pathway [120,121].

NEAT1 (Nuclear Enriched Abundant Transcript 1): Present in exosomes from CRC cells, NEAT1 could potentially encourage immunosuppressive factors and subdue anti-tumor immunity. Studies suggest that NEAT1 facilitates 5-FU chemoresistance in CRC by inducing autophagy and supporting the maintenance of stemness via the regulation of ALDH1 and c-Myc gene expression. Further, NEAT1 has been implicated in the metabolic reprogramming of recipient cells, potentially influencing mitochondrial homeostasis. The relevance of NEAT1 in tumor–immune cell interaction is highlighted by its high level of expression in exosomes derived from M2-polarized tumor-associated macrophages, which can foster immune evasion. Thus, NEAT1 could emerge as a promising therapeutic target for overcoming drug resistance and immune evasion in CRC [122,123,124].

PCAT-1 (Prostate Cancer-Associated Transcript 1): Identified in exosomes associated with various cancers, including CRC, PCAT-1 is thought to enhance the secretion of pro-inflammatory cytokines and influence cellular processes such as proliferation, invasion, migration, and apoptosis. Studies indicate that its upregulation in CRC tissues and cells can stimulate cell growth and inhibit apoptosis, primarily by targeting miR-149-5p. Thus, its dysregulation could potentially foster an inflammatory tumor microenvironment. Given its aberrant expression in diverse cancers and its influence on critical cell functions, PCAT-1 presents as a potential therapeutic target for CRC, particularly for controlling inflammation and modulating tumor–stroma immune interactions [125,126].

SNHG1 (Small Nucleolar RNA Host Gene 1): Identified in exosomes derived from CRC cells, SNHG1, a long non-coding RNA (lncRNA), suggests a potential role in intercellular communication and tumor–stroma immune interactions. Recent studies indicate that SNHG1 may promote the secretion of pro-inflammatory cytokines and the recruitment of regulatory T cells, thereby facilitating immune evasion in CRC. In addition, SNHG1 has been implicated in various CRC development processes, including cell proliferation, migration, and the epithelial–mesenchymal transition (EMT). Significantly, SNHG1 has been associated with the Wnt/β-catenin signaling pathway, a critical player in CRC progression. Furthermore, SNHG1 has been found to interact with key molecules such as MYC and SLC3A2, thereby modulating gene expression and the PI3K/AKT pathway. These findings underscore SNHG1′s potential as a crucial mediator of immune responses, inflammation, and tumor–stroma crosstalk via extracellular vesicles/exosomes in CRC [127,128,129].

SNHG14 (Small Nucleolar RNA Host Gene 14): SNHG14, a lncRNA, has been implicated in various cancer types, including CRC and hepatocellular carcinoma (HCC), demonstrating a significant impact on cancer progression. Studies have revealed that SNHG14 is upregulated in CRC, contributing to increased cell proliferation, migration, invasion, and inhibition of apoptosis. SNHG14 promotes CRC progression by negatively regulating EPHA7 through an EZH2-dependent pathway, enhancing methylation on the EPHA7 promoter, and stabilizing EZH2 mRNA by interacting with FUS and freeing it from miR-186-5p-induced silence. Additionally, SNHG14 enhances malignancy by upregulating STAT3 and increasing the proliferation and invasion of cancer cells through the downregulation of miR-613. In addition to its role in cancer progression, SNHG14 has been identified as a key regulator of immune responses within the tumor microenvironment, functioning as a microRNA sponge and impacting immune cell behavior. Through this mechanism, SNHG14 might contribute to immune evasion by inducing T cell exhaustion and modifying the tumor microenvironment. Given the observed dysregulation of SNHG14 in CRC, its involvement in tumor–stroma immune interactions via extracellular vesicles and/or exosomes is suggested, which underscores its relevance in immunity, inflammation, and tumor–stroma immune interplay. These findings highlight the potential of SNHG14 as both a therapeutic target and a biomarker for CRC and immune-related therapies [130,131,132].

SOX2OT (SOX2 Overlapping Transcript): Although the role of SOX2OT in pulmonary arterial hypertension (PAH) has been examined, its implications in CRC and the tumor microenvironment are still unexplored. Recent studies identified SOX2OT as a potential mediator of inflammation in PAH as it may promote the release of pro-inflammatory cytokines. Furthermore, SOX2OT has been linked with the transcription factor SOX2, which is known to regulate pluripotency and self-renewal of undifferentiated cells. In the context of CRC, SOX2OT has been observed to perform an oncogenic role as it is upregulated in CRC tissues and cell lines. SOX2OT silencing has been shown to suppress CRC cell proliferation, migration, and invasion in vitro and inhibit tumor growth in vivo. Mechanistically, SOX2OT acts as a competing endogenous RNA (ceRNA), sponging miR-194-5p and upregulating the expression of SOX5. These findings highlight the potential of SOX2OT as a diagnostic marker and therapeutic target in CRC, impacting tumor aggressiveness and therapy response while also modulating the tumor microenvironment and immune regulation [133,134].

TUG1 (Taurine Upregulated Gene 1): TUG1 has been identified as a significant contributor to the development of CRC through its interaction with the miR-138-5p–zinc finger E-box-binding homeobox 2 (ZEB2) axis. TUG1 is upregulated in CRC tissues and serves as a diagnostic marker for CRC, exhibiting potential as a prognostic indicator. Experimental studies using CRC cell models have shown that TUG1 promotes the epithelial-to-mesenchymal transition (EMT), which is a critical process in CRC progression. Additionally, TUG1 has been found to promote an immunosuppressive microenvironment by upregulating the expression of Siglec-15, an immune checkpoint molecule associated with HCC. The knockdown of TUG1 has been shown to reduce immunosuppression and inhibit tumor progression both in vitro and in vivo. These findings underscore the significance of TUG1 in the development of CRC and its potential as a therapeutic target for modulating the immune response and inhibiting tumor growth in CRC [135,136,137].

GAS5 (Growth Arrest Specific 5): The long noncoding RNA GAS5 has been found to be a key regulator in the context of colorectal cancer (CRC) and is known to play a vital role in the suppression of colorectal carcinogenesis. In a recent study, GAS5 was found to be commonly downregulated in CRC tissues, the serum of CRC patients, and CRC cell lines. It was further discovered that the knockdown of GAS5 led to increased CRC cell proliferation and colony formation, while its overexpression produced the opposite result. GAS5′s role in suppressing CRC was shown to be linked to the decreased expression and secretion of interleukin-10 (IL-10) and vascular endothelial growth factor (VEGF-A) via the NF-κB and Erk1/2 pathways. The neutralization of IL-10 and VEGF-A reduced tumor proliferation caused by GAS5 knockdown. The cytokines IL-10 and VEGF-A, which are inhibited by GAS5, were identified as potential targets for lncRNA-based therapies for CRC. Moreover, GAS5 was found to be markedly downregulated in a mouse model of colitis-associated cancer (CAC), and its expression showed a significant correlation with the mRNA levels of IL-10 and VEGF-A in CRC tissues. Notably, GAS5 was found to be overexpressed in M1-type macrophages compared to M2-type macrophages and was shown to block immune escape and cell proliferation by exerting the tumor-suppressive functions of M1 macrophages in CRC. These findings emphasize the significance of GAS5 as an oncosuppressive lncRNA, modulating inflammation, immune responses, and serving as a promising therapeutic target in CRC and other immune-related diseases [137,138,139,140].

Table 6 summarizes information about extracellularly released lncRNAs that can regulate immune and inflammatory interactions between tumor cells and the stroma.

As we can see, there are a number of lncRNAs that have the potential to serve as targets for therapy and prognosis markers.

## 6. Unraveling the Complexity: The Interplay of lncRNAs and Other ncRNAs in Cell-to-Cell Communication within the CRC Microenvironment

The evolving landscape of CRC research has brought non-coding RNAs, including long non-coding RNAs (lncRNAs), microRNAs (miRNAs), and circular RNAs (circRNAs), to the forefront of molecular oncology. These ncRNAs play multifaceted roles in intercellular communication within the tumor microenvironment, a crucial aspect driving cancer progression. The theory of the mRNA–lncRNA–miRNA regulatory network in CRC further emphasizes the intricate interactions in which lncRNAs act as “sponges” for miRNAs, modulating gene expression and contributing to various CRC processes [141]. This regulatory network represents a promising area for investigation, offering new perspectives for unraveling the complexity of cell-to-cell communications in CRC and potentially uncovering novel therapeutic targets [142].

Initially, the investigation of long non-coding RNAs (lncRNAs) in molecular oncology led to the perception that these molecules act primarily as suppressors or ”sponges” for the messenger RNAs (mRNAs) of corresponding genes. This simplistic understanding soon evolved into a recognition that lncRNAs could also bind microRNAs (miRNAs), preventing them from interacting with their target mRNAs. The conception that lncRNAs were merely suppressive elements started to transform into a more nuanced and intricate understanding of their role as they were found to post-transcriptionally regulate gene expression through interactions with the 3′ untranslated region (3′-UTR) of target mRNAs [142]. This transition in thinking led to the formulation of the competing endogenous RNA (ceRNA) hypothesis, which proposes that RNAs, including lncRNAs, mRNAs, and miRNAs, regulate each other’s expression by competitively binding to shared miRNA response elements (MREs). Far from being a mere suppressor, the ceRNA network was found to significantly affect CRC by influencing critical biological processes such as proliferation, differentiation, apoptosis, migration, invasion, immune response, and metastasis [143]. With the ceRNA hypothesis as a foundation, there has been a growing interest in predicting and investigating the mRNA–lncRNA–miRNA regulatory network. By unveiling the complex interactions within this network, researchers are establishing new frontiers in understanding the pathogenesis and progression of CRC.

Given the burgeoning interest in investigations of RNA–RNA networks, it is instructive to delineate a typical example of how scientists embark on this endeavor, particularly in the context of colorectal cancer (CRC). A conventional method begins with an intricate algorithm involving the collection of data via a transcriptome-level analysis and from pertinent databases such as the Gene Expression Omnibus (GEO) and The Cancer Genome Atlas (TCGA). After converting the miRNA probe IDs into mature names and identifying differentially expressed miRNAs, tools like the Limma package are employed to pinpoint dysregulated miRNAs for a further examination of their expression and prognostic value. The methodology progresses with the assembly of a lncRNA–miRNA–mRNA regulatory network, utilizing databases like TargetScan and miRDB, followed by visualization with Cytoscape software(the latest software version at the moment is 3.10.0). Subsequent gene function scrutiny is conducted with the Database for Annotation, Visualization, and Integrated Discovery (DAVID), encompassing analyses like the KEGG Pathway and GO ontology. The validation phase involves the utilization of databases like the Gene Expression Profiling Interactive Analysis (GEPIA) and PrognoScan, culminating in the construction of an innovative regulatory network, an exploration via a Gene Set Enrichment Analysis (GSEA), and a correlation assessment with variables that are valuable for cancer biology or the clinic, such as immune infiltration levels [144,145]. Examples of the results of such studies include data published in the literature on RNA interactions that can influence various aspects of colorectal cancer biology, such as immune response and immune evasion, radioresistance and treatment response, tumor occurrence and metastasis, and probiotic- and pathobiont-related ceRNA networks; most of these play general roles as prognostic indicators.

Our understanding of the immune response and immune evasion in colorectal cancer (CRC) has evolved to encompass intricate networks involving noncoding RNAs. Among these, long noncoding RNAs (lncRNAs) such as LINC00657 and NEAT1 have emerged as crucial regulators. LINC00657 affects CD8+ T cells’ function and infiltration, which are vital components of the cancer immune response, by inhibiting the cells’ cytotoxicity and augmenting the expression of CD155, a tumor marker linked to immune suppression. This RNA networking phenomenon is further corroborated by NEAT1′s modulation of tumorigenesis-related pathways by sponging various miRNAs [146,147,148]. The regulatory mechanisms of LINC00657 and NEAT1 underscore their significant roles in altering the immune response.

Immune evasion is further delineated through networks involving specific lncRNAs, such as the H19/miR-29b-3p/PGRN axis and SNHG7. The former network fosters the epithelial–mesenchymal transition (EMT) in CRC, thereby influencing the Wnt signaling pathway, while the latter acts as a competing endogenous RNA (ceRNA) to promote proliferation and liver metastasis [149]. Such RNA-based interactions may furnish insights into immune modulation within the tumor microenvironment.

Additional networks like CECR7-miR-206/miR-107, which may influence cytotoxic T cells and CRC invasion [150], and the ceRNA regulatory network involving NEAT1, XIST, and hsa-miR-195-5p [144], further exemplify how RNA networks are intertwined with immune responses in CRC. These interactions are significant for the infiltration of immune cells, such as CD4+/CD8+ T cells and macrophages, and immune escape in the CRC tumor microenvironment (TME).

The interaction of noncoding RNAs in radioresistance and treatment response represents another critical area. The lncRNA SP100-AS1, identified as pivotal in CRC radioresistance and upregulated in radioresistant CRC tissues, interacts with ATG3 protein and sponges miR-622, affecting autophagic activity and possibly contributing to radioresistance. The SP100-AS1/miR-622/ATG3 axis might be targeted to enhance the efficacy of radiation therapy in CRC, signifying its importance in controlling radioresistant tumor cells [151]. Further investigations into additional RNA networks involved in radioresistance may provide more insights into regulating treatment responses in cancer.

In other contexts, various non-coding RNAs have been implicated in tumor occurrence and metastasis in CRC, orchestrating numerous cellular biological activities and signaling pathways. One focused study identified specific regulatory networks involving two upregulated microRNAs (the miRNAs miR-141 and miR-216a), three lncRNAs (ARHGEF26-AS1, AP004609.1, and LINC00491), and three messenger RNAs (the mRNAs TPM2, FJX1, and ULBP2) associated with clinical outcomes, cell invasion, and key roles in the PI3K/AKT and Wnt signaling pathways. MiR-216a was further confirmed as an independent prognostic factor for CRC [152].

In separate examples, networks such as the lncRNA H19/miR-29b-3p/PGRN axis promote the EMT in CRC, enhancing migration and invasion through the Wnt and MAPK signaling pathways [149]. Other networks, like the lncRNA SNHG7 acting as ceRNA, foster proliferation and liver metastasis by regulating the PI3K/Akt/mTOR pathway [149]. Certain miRNA networks, including hsa-miR-1827-FOXP2 and hsa-miR-448-ORC6, which are linked to invasiveness, drug resistance, and tumor suppression in CRC, have been identified through computational methods and warrant further experimental validation [153].

Furthermore, the DANCR network’s role in CRC metastasis, affecting cell migration, invasion, angiogenesis, and immune cell reprogramming through the DANCR/microRNA-518a-3p/MDMA and DANCR/miR-185-5p/HMGA2 axes [150] and the lncRNA CECR7 via the CECR7-miR-206/miR-107 network, potentially impacting immune modulation and extracellular matrix interaction within the TME [150], is noteworthy.

Collectively, these ncRNA networks symbolize intricate interplays governing tumor occurrence, metastasis, and the surrounding microenvironment in CRC.

In the probiotic-related ceRNA network, 75 lncRNAs have been identified to interact with 8 miRNAs, including hsa-mir-429, hsa-mir-141, hsa-mir-140, hsa-mir-22, hsa-mir-132, hsa-mir-454, hsa-mir-153, and hsa-mir-143. This complex network provides promising insights into probiotics’ potential to reverse gene expression in tumor cells, countering CRC carcinogenesis and progression [154].

Conversely, the pathobiont-related ceRNA network, consisting of 49 lncRNAs and 4 miRNAs (hsa-mir-223, hsa-mir-32, hsa-mir-96, and hsa-mir-106a), seemingly promotes the expression of oncogenes or inhibits tumor suppressors, thus facilitating the progression of CRC. It underscores the potential role of pathogenic bacteria in exacerbating the disease, with functional annotation showing enrichment in cellular metabolic regulation and the p53 signaling pathway, indicating that microbiota may mainly engage in CRC development through these processes [152,154].

An intriguing area is the exosomal competing endogenous RNA (ceRNA) network, comprising a multifaceted interplay between mRNAs, lncRNAs, and miRNAs. One study unveiled an exosomal ceRNA regulatory network including 40 lncRNAs, 2 miRNAs, and 5 mRNAs, identifying two exosomal regulatory axes: lncRNA G016261-miR-150-5p-RBM48 and lncRNA XLOC_011677-miR-10b-5p-BEND3. The differential expression of these key molecules may affect CRC carcinogenesis and development by modulating essential cellular pathways. These insights into the exosomal ceRNA network’s molecular regulation pathways may contribute to a better understanding of the pathogenesis of CRC and potentially lead to the discovery of novel diagnostic or therapeutic targets [155].

In conclusion, the exploration of noncoding RNA networks in colorectal cancer (CRC) has uncovered a complex and nuanced landscape governing various aspects of tumor biology, including immune evasion, metastasis, and treatment response. The intricate interplay involving lncRNAs such as LINC00657, NEAT1, H19, and SNHG7, as well as multiple miRNAs, has emphasized their pivotal roles in the modulation of the immune response, cellular pathways like Wnt, MAPK, and PI3K/Akt/mTOR, and the progression of CRC. Probiotic- and pathobiont-related ceRNA networks further illustrate the potential influence of the microbiota in CRC, emphasizing the need to investigate their functional implications. Moreover, insights into radioresistance mediated by lncRNA SP100-AS1 and the discovery of exosomal ceRNA networks provide promising avenues for therapeutic interventions and novel diagnostic markers. These findings collectively underline the paramount significance of noncoding RNA networks in CRC, offering a profound understanding that can guide future research and clinical applications in oncology.

## 7. Exploring Strategies for Manipulating lncRNAs to Enhance Anti-Tumor Immunity in CRC Patients

The escalating exploration of long non-coding RNAs (lncRNAs) within the cancer microenvironment has emerged as fertile ground for pioneering new cancer therapies. This section will meticulously examine an array of strategies that capitalize on the distinct properties of lncRNAs to bolster anti-tumor immunity. Recognizing the inherent risks linked with therapeutic interventions directed at intracellular lncRNAs, our focus shifts towards more promising and potentially safer modalities. These may encompass targeting lncRNAs that are prolifically released extracellularly or the innovative utilization of artificial microvesicles as therapeutic vectors.

### 7.1. Potential Methodologies for Manipulating Extracellularly Released lncRNAs

Employing EV inhibitors and modulating uptake: influencing the microvesicular release, uptake, and subsequent impact on the non-coding RNA content within these vesicles. Potential methodologies include employing EV inhibitors and modulating their uptake. Certain pharmacological inhibitors, such as GW4869, can limit the biogenesis or release of extracellular vesicles (EVs), thereby reducing the number of lncRNA-packed vesicles in the tumor microenvironment. GW4869 has demonstrated potential in inhibiting processes like the epithelial–mesenchymal transition in non-small-cell lung cancer (NSCLC), particularly in those cases with EGFR mutations [156]. However, caution is warranted as such inhibitors might also impact essential physiological processes and could potentially induce off-target effects [157]. Concurrently, an intriguing approach involves the manipulation of the uptake of EVs by recipient cells. Gaining insights into and manipulating the mechanisms through which EVs and their lncRNA cargo are absorbed by cells could potentially control the impact of these lncRNAs. Selectively inhibiting the uptake of EVs by tumor cells without affecting normal cells, however, presents a significant challenge. This approach could potentially lead to unintended effects on healthy tissues [158,159]. This concept is schematically depicted in Figure 2.

Genetic modification: advanced gene-editing tools such as CRISPR-Cas9 and RNA interference (RNAi) have the potential to specifically target genes encoding lncRNAs within colorectal tumor cells. For instance, studies have revealed that the knockdown of lncRNA MALAT1, H19, NEAT1, and TUG1, among others, can influence cellular processes, potentially altering the course of the progression of CRC [160,161,162,163,163]. The application of RNAi to NEAT1 not only affected immune cell differentiation but also showed a potential influence on the STAT3 protein, a critical molecule in inflammatory responses [162]. Meanwhile, silencing TUG1 was found to enhance radiosensitivity in prostate cancer, suggesting possible implications for enhancing the effectiveness of radiation therapy in CRC [163]. Speculatively, genetically engineered immune cells, such as chimeric antigen receptor (CAR) macrophages with modified lncRNA expression, might offer a novel approach to optimizing their anti-tumor activity and navigating the immunosuppressive tumor microenvironment [163]. It may also be possible to modify lncRNA genes or genes of their effectors. This concept is presented in Figure 3.

However, while these techniques have immense potential, their in vivo application in humans still encounters significant challenges, including issues related to delivery and potential off-target effects. Furthermore, ethical and regulatory considerations surrounding genetic modification in humans also form an essential part of the development and implementation of these therapeutic strategies.

The use of lncRNA sponges: the creation of synthetic lncRNA sponges and RNA interference presents a fascinating possibility. These could capture and neutralize specific lncRNAs once they are released from EVs. Although this strategy does not directly interact with vesicle-contained lncRNAs, it does target them post release, mitigating their functional impact within the tumor microenvironment [164,165]. While a promising approach, there might be challenges related to the efficient delivery of these molecules to the tumor site. Additionally, the risk of off-target effects exists if the sponge binds to and neutralizes other RNAs. To a large extent, these questions might be solved with the following approach we want to mention: targeted drug delivery with exosomes (Figure 4).

Targeted drug delivery: targeted drug delivery continues to present promising advancements, particularly in the field of CRC treatment. Central to these developments is the utilization of exosomes as carriers for the specific delivery of therapeutic agents. This sophisticated approach involves encapsulating antisense oligonucleotides (ASOs) or small interfering RNAs (siRNAs) into EVs and subsequently releasing them to inhibit target lncRNAs specifically. However, such a method requires the intricate manipulation of EVs and presents technical challenges, including potential safety concerns linked to modifying EVs and using them as drug delivery platforms. These could contribute to unforeseen side effects [166,167]. Illustrating this approach, one study effectively utilized autologous serum-derived exosomes to carry siRNAs, leading to a reduction in lung metastases in melanoma [168]. Similarly, another research capitalized on the inherent characteristics of plant-derived exosomes, packaging a range of tumor necrosis factors, small-molecule drugs, and siRNAs for targeted treatment. This use of plant-derived exosomes suggests potential for both large-scale production and economic feasibility [169]. In addition to these techniques, lncRNAs such as H19, often aberrantly expressed in various cancers including CRC, can be targeted. Synthetic anti-H19 constructs can be packed into small extracellular vesicles to suppress tumor growth and metastasis. This method has proven to be non-toxic, non-immunogenic, and biocompatible [170]. Furthermore, RNA interference (RNAi) techniques are being investigated, with nanoparticles acting as carriers for siRNA to hinder the rampant growth and proliferation of cancer cells. Some of these approaches have shown promising antitumor activities in clinical trials, further solidifying the potential role of exosome- and nanoparticle-based targeted drug delivery in CRC treatment [171].

### 7.2. Immune Processes That We Can Regulate via lncRNAs

Altering immunogenic protein expression via lncRNAs: common CRC treatments encompass surgery, radiation, and chemotherapy. Recently, immunotherapy has emerged as a potential addition to this standard regimen. This development underscores the need to understand and investigate lncRNAs as they regulate the expression of immunogenic proteins or tumor-associated antigens [172]. This regulation could stimulate an anti-tumor immune response, enhancing the effectiveness of current immunotherapies.

CCAT1 is a critical lncRNA in this context; it is found in the EVs of CRC cells and is typically overexpressed in CRC. This overexpression allows CCAT1 to alter tumor antigen production by controlling gene expression, acting as a microRNA sponge, and modulating signaling pathways. Consequently, increasing the visibility of CRC cells to the immune system could enhance the efficacy of existing immunotherapies [173].

In a further exploration of non-coding genome regulation, protein arginine methyltransferase 5 (PRMT5) and the transcription regulator E2F1, both of which are overexpressed in many cancers, significantly influence lncRNA gene expression, leading to peptides associated with MHC class I proteins. By modifying this expression, these lncRNA-derived peptides could trigger a robust immune response, potentially delaying tumor growth and representing a potentially powerful approach for controlling tumor immunogenicity [174].

The tumor suppressor FENDRR (fetal-lethal non-coding developmental regulatory RNA, LncRNA FOXF1-AS1) is another critical lncRNA. FENDRR’s expression is linked to the epigenetic modulation of genes involved in tumor immunity. Tumor cells that typically exhibit increased immune activation and MHC class I molecule expression also show heightened FENDRR expression levels. Modifying FENDRR expression appears to enhance inflammatory and WNT signaling pathways in tumors, highlighting its potential role in shaping immune-relevant phenotypes in tumors [175].

The lncRNA LIMIT (lncRNA inducing MHC-I and tumor immunogenicity tumor), which is identifiable in humans and mice, enhances the activity of the guanylate-binding protein (GBP) gene cluster. This upregulation disrupts the association between HSP90 and heat shock factor-1 (HSF1), leading to the transcription of MHC-I machinery, which, in turn, enhances tumor immunogenicity and the effectiveness of checkpoint therapy [176].

In summary, these findings underscore the potential of lncRNAs for enhancing cancer immunotherapies, representing a promising avenue for future therapeutic strategy development.

The lncRNA-mediated regulation of immune checkpoints: immunotherapy, particularly when targeting the PD-1/PD-L1 immune checkpoint, significantly enhances survival outcomes in various cancers. Understanding the molecular basis of therapeutic responses to these treatments is critical for identifying suitable patient candidates [131]. An investigation into lncRNA expression in patients undergoing anti-PD-1/PD-L1 immunotherapy identified lncRNA NEAT1. Its upregulation was common among patients with melanoma who exhibited a complete therapeutic response and patients with glioblastoma (GBM) who demonstrated longer survival. NEAT1′s expression was closely associated with IFNγ pathways and the downregulation of cell-cycle-related genes [177].

The lncRNA HOTAIR, found in the EVs of CRC cells, potentially plays a significant role in modulating immune checkpoints. HOTAIR, known for its intricate role in chromatin dynamics and cell cycle regulation, interacts with immune checkpoints by modulating the expression of proteins such as PD-L1. In addition to a notable regulatory role in osteogenesis and cell senescence, recent studies have indicated that tumor-derived HOTAIR can polarize B cells towards a regulatory phenotype marked by programmed cell death-ligand 1 (PDL1) in CRC, subsequently inducing PDL1+ B cells to suppress CD8+ T cell activity [178].

Moreover, were identified six hypoxia-immune-related lncRNAs, namely, ZNF667-AS1, LINC01354, LINC00996, DANCR, CECR7, and LINC01116, which can predict CRC survival and sensitivity to immunotherapy [150]. Also, lncRNA SNHG4 modulates MET by sponging miR-144-3p, participating in the malignant biological behaviors and immune escape of CRC [179]. In addition, lncRNA KCNQ1OT1, which is overexpressed in tumor tissues and tumor cell-derived exosomes, could regulate PD-L1 ubiquitination via the miR-30a-5p/USP22 pathway, thereby promoting CRC immune escape [180].

Collectively, these findings underscore the potential of lncRNAs, such as HOTAIR, NEAT1, and KCNQ1OT1, to influence immune checkpoints and improve the efficacy of immunotherapy in CRC patients. However, the verification of this potential requires further large-scale, long-term follow-up studies.

Modifying the TME via lncRNA manipulation: the tumor microenvironment (TME) significantly influences the progression of CRC, affecting both the tumor’s behavior and the body’s response. The LncRNA MALAT1 can alter the TME in the progression of colorectal cancer CRC (CRC) by promoting angiogenesis and a pro-inflammatory environment. It accomplishes this by binding with miR-15 family members and SFPQ, thereby enhancing β-catenin signaling and releasing PTBP2, which escalates both the transcriptional and translational levels of the RUNX2 gene [36].

Moreover, other lncRNAs are crucial in the progression of CRC. For instance, LINC00659 is encapsulated in exosomes derived from cancer-associated fibroblasts (CAFs), and its increased expression in CAF-derived exosomes significantly promotes CRC cell proliferation, migration, and invasion [181]. Similarly, LINC00543, which is overexpressed in CRC tissues, facilitates CRC metastasis by enhancing the epithelial–mesenchymal transition (EMT) and remodeling the TME [182].

Furthermore, oxidative stress-related lncRNAs, including AC034213.1, AC008124.1, LINC01836, USP30-AS1, AP003555.1, AC083906.3, AC008494.3, AC009549.1, and AP006621.3, have been identified as playing critical roles in the progression of CRC, offering new avenues for prognosis prediction [183]. Lastly, CAF-derived exosomal WEE2-AS1 facilitates the progression of CRC by promoting MOB1A degradation to inhibit the Hippo pathway [184].

Considering their significant roles in the progression of CRC, lncRNAs offer promising avenues for therapeutic intervention, allowing for disruption of the supportive environment tumors create for their own growth. Furthermore, these lncRNAs could serve as valuable biomarkers for predicting the recurrence and metastasis of CRC, facilitating improved diagnosis and prognosis. Therefore, further study on the roles of lncRNAs in the TME of CRC could lead to significant advancements in the understanding and treatment of this disease.

Utilizing lncRNAs as vaccine adjuvants: recent discoveries have highlighted the significant role that lncRNAs play in immunology. This understanding could be exploited for advanced vaccine design, particularly for CRC. LncRNAs are crucial in regulating immune cell differentiation and activation, thereby influencing both innate and adaptive immunity [185].

Multiple studies have elucidated the impact of lncRNAs on the immune response to infections, vaccinations, and inflammation, among other biological processes. In particular, lncRNAs such as LUCAT1 and MALAT1 have been associated with modulating immune responses, and changes in their expression levels have been observed in vaccination breakthrough (VBT) infections [186,187].

For instance, a transcriptome sequencing study on VBT and unvaccinated COVID-19 patients disclosed distinct lncRNA and mRNA expression patterns among VBT patients [187,188]. This pattern included a downregulation of lncRNAs associated with immune and inflammatory responses. Interestingly, this lncRNA-mediated modulation may explain the occurrence of breakthrough infections with milder symptoms, indicating a potential role for lncRNAs in regulating the body’s immune response following vaccination.

Vaccine trials involving influenza and yellow fever have also identified differentially expressed lncRNAs associated with post-vaccination immune responses, suggesting that lncRNAs may be linked to immune regulation. For example, the lncRNA MIAT consistently showed downregulation one day after an influenza vaccination across multiple cohorts, suggesting a potential role in the early immune response to vaccines. Similarly, the lncRNA DANCR was upregulated in vaccine responders, hinting at a role in regulating the differentiation of antibody-secreting cells [188].

Importantly, this knowledge can be leveraged for CRC treatment. Certain lncRNAs, such as MALAT1, play key roles in the progression of CRC by modulating angiogenesis and fostering a pro-inflammatory environment [37,161,181]. By manipulating these lncRNAs, we could potentially enhance the efficacy of anti-cancer vaccines by altering the tumor microenvironment to be less hospitable to cancer cells, thereby improving patient outcomes.

It is noteworthy that the current therapeutic potential of targeting the immune system in colorectal cancer (CRC) through long noncoding RNAs (lncRNAs) is virtually unexplored. However, several clinical trials are shedding light on the significance of lncRNAs in CRC diagnostics and prognosis.

The clinical trials with IDs NCT04729855 and NCT04269746 emphasize the role of specific lncRNAs such as HOTTIP and CCAT1 in CRC, investigating their potential as less-invasive prognostic markers and exploring their correlation with susceptibility to chemotherapy. For example, the trial “Association of Autophagy-related Genes, LncRNA, and SNPs With Colorectal Cancer in the Egyptian Population” examined the expression levels of HOTTIP in peripheral blood mononuclear cells, correlating them with the progression of CRC and resistance to therapy. Additionally, the study titled “Assessment Of Long Noncoding RNA CCAT1 Using Real Time-Polymerase Chain Reaction In Colorectal Cancer Patients” explored CCAT1′s diagnostic and clinical utility, comparing it against traditional markers and assessing its relation to tumor staging.

Although not directly related to lncRNAs, a Phase 1 clinical trial (NCT04660929) focusing on the treatment of HER2-overexpressing solid tumors (including CRC) using CAR macrophages hints at a broader connection between lncRNAs and cancer biology. The study, while not explicitly dealing with lncRNAs, introduces the concept of HER2 regulation and immune response mechanisms, thereby offering insights into potential targets for precise therapeutic interventions. Understanding lncRNA-mediated regulation in this context could unlock innovative avenues for treating CRC and other malignancies, fostering the development of novel therapeutic and prognostic strategies.

Despite this promising potential, it should be noted that further research is required. Understanding the role of lncRNAs in the immune response to vaccination is an emerging field, and there is still much to be discovered. Future studies should focus on validating these findings, unraveling the precise mechanisms of lncRNA-mediated immune regulation, and exploring their potential use as vaccine adjuvants, specifically for CRC.

## 8. Conclusions: Future Perspectives on Long Non-Coding RNAs in CRC

The pivotal role of lncRNAs in the pathogenesis of CRC, immunity, and the intercellular communication within the tumor microenvironment of CRC is becoming increasingly clear. Acting as orchestrators of gene regulation, lncRNAs can modulate cellular processes, shaping tumor biology and influencing patient responses to therapy. As we delve further into the exploration of lncRNAs, we unearth a complex and rich matrix of intertwined mechanisms, offering promising avenues for decoding the intricacies of CRC.

We stand on the brink of an emerging paradigm in our understanding of CRC, evolving from a cell-centric perspective to a network-focused model. This model posits lncRNAs as essential components of sophisticated intercellular communication networks, potentially blurring the boundaries between cells and tissues. lncRNAs do not function in isolation; they participate in expansive networks of interactions, possibly involving other types of RNA, such as microRNAs (miRNAs) and messenger RNAs (mRNAs). The intricate web they weave within the cellular environment may provide novel insights into the molecular underpinnings of CRC, thereby unveiling new therapeutic opportunities.

Furthermore, the intersection between the gut microbiome and lncRNAs presents a fertile ground for additional investigation. Given the established roles of lncRNAs in cellular communication and immune modulation, they could be perceived as critical players in the complex dynamic between gut microbiota and host immunity. Unraveling the potential crosstalk between lncRNAs and the microbiome could present exciting opportunities for enhancing our understanding of CRC and its management [189].

The influence of lncRNAs also extends to the realm of drug resistance in CRC. Emerging evidence points towards the involvement of lncRNAs in the development of resistance to chemotherapeutic agents such as 5-fluorouracil (5-FU) and oxaliplatin. For example, the lncRNA H19 has been implicated in resistance to 5-FU. A deeper exploration of the roles of lncRNAs in drug resistance could not only assist in predicting patient responses to specific treatments but also provide strategies for overcoming such resistance [190].

The distinct expression patterns of lncRNAs in various CRC subtypes spark intriguing possibilities for future interventions. One such proposition is the development of “lncRNA vaccines.” This innovative approach aims to stimulate the immune response against cells that express specific lncRNAs associated with tumor progression or immune evasion, potentially introducing a new frontier in cancer therapy.

Moreover, the diversity of lncRNA expression within different cells of the same tumor suggests a dynamic interplay. It prompts questions about how lncRNA variability might influence the evolutionary trajectory of CRC. This line of inquiry could yield valuable insights into intratumoral heterogeneity, clonal evolution, and the emergence of therapy-resistant clones.

The future exploration of lncRNAs promises transformative shifts in our approach to CRC research, diagnosis, and treatment. The journey ahead is challenging, yet every hurdle crossed brings us a step closer to the ultimate goal: a world that has overcome the burden of CRC.

## Figures and Tables

**Figure 1 biomedicines-11-02411-f001:**
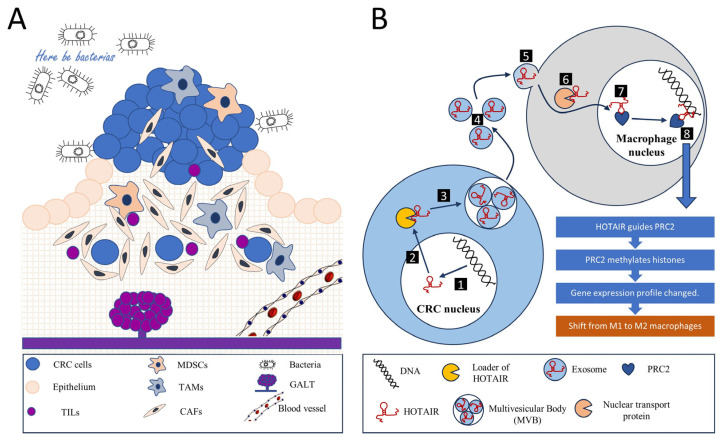
Tumor microenvironment and intercellular communications. (**A**) A colorectal cancer tumor is a complex, multicellular, and multi-dimensional entity. The tumor microenvironment encompasses not only tumor cells originating from the intestinal epithelium but also various components that help regulate metabolism, nutrition, and immune responses. These components include an extracellular matrix, a diverse population of tumor-infiltrating lymphocytes (TILs), myeloid-derived suppressor cells (MDSCs), tumor-associated macrophages (TAMs), and cancer-associated fibroblasts (CAFs), along with blood and lymphatic vessels. Another characteristic aspect of colorectal cancer is the presence of gut-associated lymphoid tissue (GALT). Additionally, bacteria play an essential role within the microenvironment. All these components interact to maintain a balance, thereby establishing a conducive environment for tumor growth. Among other forms of interaction, intercellular communication involving long non-coding RNAs (lncRNAs) is noteworthy. (**B**) The signaling mechanism of the long non-coding RNA (lncRNA) HOTAIR can serve as an example of intercellular communication through an lncRNA. In the nucleus of the tumor cell, the transcription and primary modification of the HOTAIR gene RNA occur (1), after which the mature lncRNA HOTAIR moves to the cytoplasm (2). Here, via a loader mechanism, HOTAIR is incorporated into a multivesicular body (MVB) where exosome particles loaded with HOTAIR are formed (3). These exosomes are then released into the extracellular environment (4). Their content is subsequently absorbed by macrophages via pinocytosis (5). Nuclear transport proteins guide HOTAIR into the macrophage’s nucleus (6), where this RNA interacts with the Polycomb Repressive Complex 2 (PRC2) chromatin remodeling complex (7). The HOTAIR-PRC2 complex identifies specific genomic regions (8), leading the PRC2 complex to methylate histones and silence a series of genes, thereby altering the gene expression profile of the macrophage. This results in a switch in macrophage polarization from M1 (anti-tumor) to M2 (pro-tumor), helping to sustain an environment conducive to tumor growth.

**Figure 2 biomedicines-11-02411-f002:**
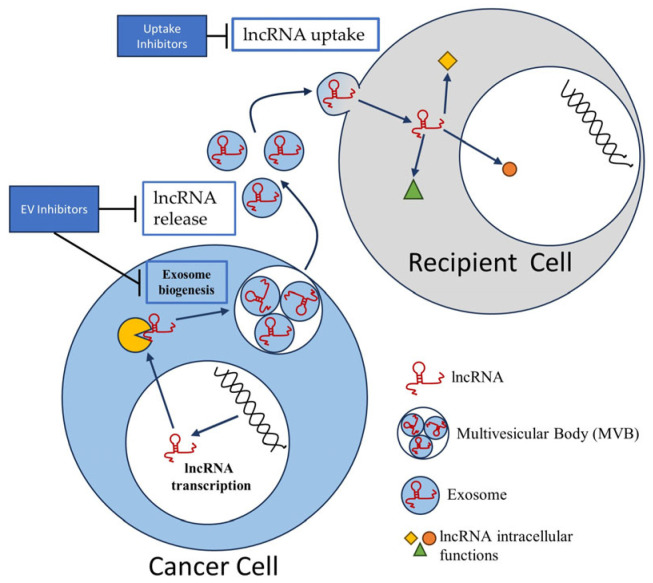
Extracellular vesicle biogenesis and lncRNA uptake: inhibition strategies in cancer Cells. This illustration provides a view of the interaction between a cancer cell and a recipient cell, focusing on extracellular vesicles (EVs) and long non-coding RNAs (lncRNAs). Within the cancer cell, the processes of transcription, multivesicular body (MVB) formation, and exosome biogenesis are depicted, leading to the release of lncRNA-packed exosomes into the extracellular environment. A rectangle marked “EV Inhibitors” signifies the inhibition of exosome biogenesis and release. The recipient cell is shown absorbing the lncRNAs contained within the exosomes, a process that is represented as being inhibited by a rectangle marked “Uptake Inhibitors”. The entire figure integrates these components to illustrate the complex mechanisms of EV formation, release, and uptake, and the potential therapeutic strategies of inhibiting these processes.

**Figure 3 biomedicines-11-02411-f003:**
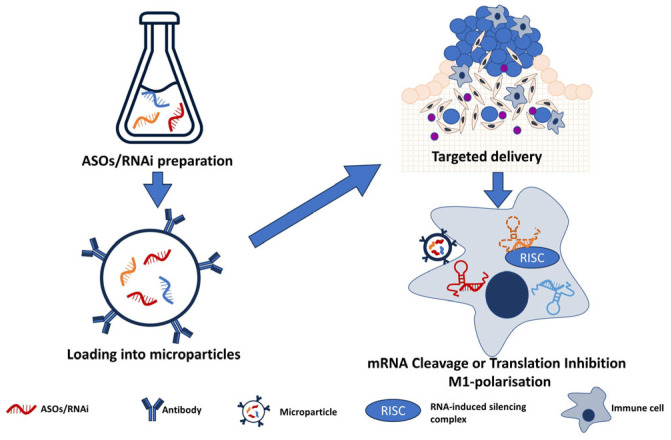
Principles of ASOs/RNAi for lncRNA silencing in colorectal cancer tumor cells. The illustration offers a step-by-step visual guide of the process involving ASOs/RNAi-based gene silencing, beginning from the in vitro production of therapeutic molecules to their eventual interaction and impact within colorectal cancer tumor cells. “ASOs/RNAi Preparation”: displayed as a lab flask containing miniature, colored strands of DNA or RNA, this section represents the synthesis of antisense oligonucleotides (ASOs) or RNA interference molecules (RNAi). “Loading into Microparticles”: this stage is portrayed by these small strands being encapsulated into larger, sphere-like structures. This represents the loading of ASOs/RNAi into microvesicles or microparticles. Additionally, these microparticles could be engineered to carry a specific antibody to facilitate targeted binding to specific cells. “Targeted Delivery”: delivery poses a significant challenge in this process. The microparticles carrying the therapeutic molecules are depicted entering the complex cellular environment of a colorectal tumor, which is composed of a myriad of different cell types. “mRNA Cleavage or Translation Inhibition and M1-Polarisation”: within the tumor cells, the microparticles discharge the ASOs or RNAi. These molecules are illustrated as binding to their long non-coding RNA (lncRNA) targets (depicted as strands of varying colors within the cells), leading to the degradation of the lncRNA, effectively “silencing” them. This step includes the RNA-induced silencing complex (RISC) mechanism, which allows for either mRNA cleavage or translation inhibition, leading to the silencing of the target genes.

**Figure 4 biomedicines-11-02411-f004:**
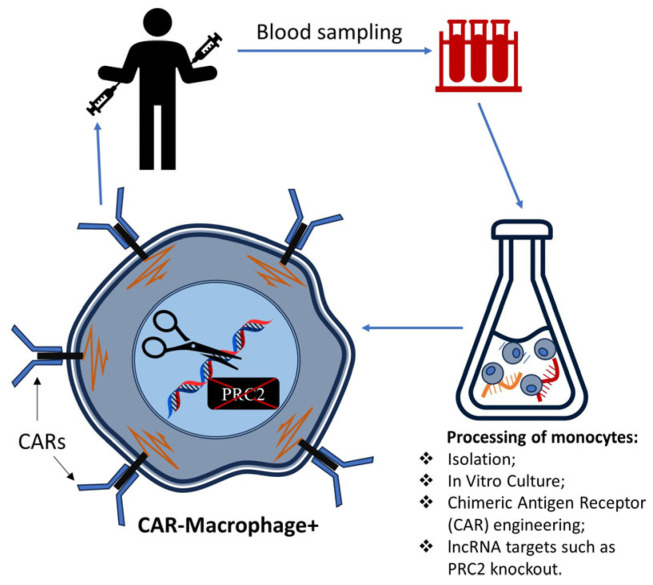
Conceptual model of engineering CAR macrophages through lncRNA HOTAIR targeting and PRC2 knockout for cancer immunotherapy. The illustration outlines a hypothetical framework for enhancing cancer immunotherapy through genetic engineering. It begins with the isolation of a monocyte from a patient’s blood sample for in vitro cultivation. Within this controlled setting, Chimeric Antigen Receptor (CAR) structures are hypothetically engineered onto the monocyte. Following this, a targeted knockout of the protein PRC2, which is associated with lncRNA HOTAIR, is performed. This theoretical modification could transform the monocyte into a CAR macrophage, a change depicted by a clear alteration in the cell’s appearance. The final stage of the model demonstrates the potential reintroduction of the enhanced CAR macrophage into the patient’s body, encapsulating an innovative concept for the future development of cancer treatment strategies.

**Table 1 biomedicines-11-02411-t001:** The classification of long non-coding RNAs (lncRNAs) according to the genomic locations of Their genes.

Classification	Description
Sense lncRNAs	Transcribed from the same strand as a protein-coding gene and may overlap entirely or partially with the gene [12].
Antisense lncRNAs	Transcribed from the opposite strand of a protein-coding gene and may overlap with exons or introns [13].
Intronic lncRNAs	Located within the introns of a protein-coding gene but transcribed independently [14].
Intergenic lncRNAs	Situated between protein-coding genes and do not overlap with them. Also known as long intergenic non-coding RNAs (lincRNAs) [15].
Bidirectional lncRNAs	Transcribed in close proximity to a protein-coding gene but in the opposite direction [16].
Enhancer lncRNAs (eRNAs)	Associated with enhancer regions and may regulate the activity of enhancers, influencing gene expression [17].

**Table 2 biomedicines-11-02411-t002:** Overview of long non-coding RNA (lncRNA) functions and mechanisms in cellular processes.

Broad Function	Specific Mechanism	Description
Gene Expression Regulation	Transcriptional Control	Involves the activation/repression of transcription, enhancer activity, RNA polymerase interference, chromatin remodeling, histone modification, and DNA methylation [18,19,20].
	Post-transcriptional Control	Includes the regulation of splicing, mRNA stability, and translation [21,22].
RNA Interactions	miRNA Sponging	lncRNAs may sequester miRNAs away from their target mRNAs [23].
	RNA-RNA Interactions	Includes base pairing with other RNAs, affecting function or stability [24].
RNA–Protein Interactions	Scaffolding and Sequestration	lncRNAs can act as scaffolds for protein complexes or sequester proteins away from functional locations. May overlap with gene expression regulation and RNA interactions [25,26].
Structural Roles	Nuclear Architecture	Contributes to the organization of nuclear structures. May have indirect effects on gene regulation [27].
Signaling Regulation	Pathway Modulation	Involves interactions with signaling molecules or pathway components, potentially impacting various cellular processes, including gene expression, growth, and stress [28,29].

**Table 3 biomedicines-11-02411-t003:** Typical molecular genetic alterations associated with the CMSs of CRC.

CMS Subtype	Typical Molecular Genetic Alterations
CMS1 (MSI Immune)	High microsatellite instability (MSI-H), DNA mismatch repair (MMR) deficiency, hypermutated phenotype, and a high neoantigen load.
CMS2 (Canonical)	Chromosomal instability, a high level of somatic copy number alterations, the activation of the WNT and MYC signaling pathways, and mutations in APC and TP53.
CMS3 (Metabolic)	Microsatellite stable (MSS), metabolic dysregulation, KRAS mutations, and involvement in the PI3K/AKT signaling pathway and possibly others that affect metabolism, such as CDK2 signaling.
CMS4 (Mesenchymal)	Stromal invasion and involvement in the RAS/MAPK, Rb/E2F, CDK8/β-catenin, and Raf/ERK pathways. There is a focus on the epithelial-to-mesenchymal transition (EMT) and the regulation of pathways related to cell growth and migration.

**Table 4 biomedicines-11-02411-t004:** Interactions and functions of lncRNAs in CRC CMSs.

lncRNA	CMS Subtype	Main Characteristics	References
HOTAIR	CMS1 (MSI Immune)	Antisense lncRNA. Gene expression regulation. Oncogenic. Interacts with PRC2 and LSD1 to modulate H3K27 methylation, affecting gene silencing. Consequences: PTEN methylation, PI3K/p-AKT/p-MDM2/p53, and PI3K/AKT/mTOR pathways in tumorigenesis; regulates ASTN1, PCDHA1, and MUC5AC in metastasis.	[49,50,53,54]
LINK-A (LINC01139)	CMS1 (MSI Immune)	Intergenic lncRNA. Pathway modulation. Oncogenic. Facilitates crosstalk between the PIP3 and GPCR pathways, attenuating PKA activity on TRIM71, leading to the degradation of PLC and tumor suppressors Rb and p53. Directly binds to phosphatidylcholine, AKT, and PIP3, causing AKT hyperactivation and tumorigenesis.	[55,56,57,58]
CCAT1	CMS2 (Canonical)	Intergenic lncRNA. Nuclear architecture; scaffolding. Oncogenic. Mediates chromosome looping with CTCF, affecting c-Myc promoter and promoting c-Myc expression. Acts as a ceRNA; serves as a scaffold for epigenetic complexes, with chromosome looping central to interaction.	[59,60]
CRNDE	CMS2 (Canonical)	Intergenic lncRNA. miRNA sponging; pathway modulation; transcriptional control. Oncogenic. Molecular sponge for miRNAs; promotes cell growth. Activates/inhibits the Wnt/β-catenin, PI3K/AKT/mTOR, Ras/MAPK, and Notch1 signaling pathways. Binds to EZH2.	[61,62,63]
lncRNA-ATB	CMS3 (Metabolic)	Intergenic lncRNA. miRNA sponging. Oncogenic. Interacts with miR-141-3p and miR-200c, influencing the CDK2 pathway, affecting EMT process, and contributing to cancer progression.	[64,65,66,67]
RP11-462C24.1 (RPL34-DT)	CMS3 (Metabolic)	Intergenic lncRNA. Pathway modulation; transcriptional control. Oncosuppressive. Upregulates HSP70; inhibits the PI3K/AKT signaling pathway.	[68,69]
H19	CMS4 (Mesenchymal)	Intergenic lncRNA. RNA interactions; pathway modulation. Oncogenic. Promotes CRC progression by targeting RB with miR-675, sponging miR-200a and miR-138, leading to HMGA2 upregulation. Activates the RAS/MAPK, Rb/E2F, CDK8/β-catenin, and Raf/ERK pathways.	[70,71,72,73,74]
lincRNA-p21 (TP53COR1)	CMS4 (Mesenchymal)	Intergenic lncRNA. RNA–RNA interactions; RNA–protein interactions. Oncosuppressive. Interacts with the JUNB and CTNNB1 mRNAs, reducing translation. Antagonism via HuR. mTOR/lincRNA-p21 involved in carcinogenesis, progression, metastasis. Part of the p53 network.	[75,76,77,78]

**Table 5 biomedicines-11-02411-t005:** Mechanisms of innate and adaptive immune interactions in the CRC tumor–stroma microenvironment.

Mechanism	Key Components	Effect	Key Details
The secretion of pro-inflammatory cytokines	CAFs and TAMs	Induces inflammation	CAFs and TAMs secrete IL-6 and TNF-α, which can foster chronic inflammation, paradoxically promoting tumor progression.
The secretion of immunosuppressive factors	CAFs and TAMs	Immunosuppression	CAFs and TAMs secrete TGF-β, IL-10, and PD-L1, which inhibit T cells and promote Tregs.
T cell exhaustion	T cells	Immunosuppression	Chronic exposure to tumor antigens and inflammatory signals can lead to a state of T cell dysfunction characterized by sustained expression of inhibitory receptors (PD-1 and CTLA-4).
The recruitment of regulatory immune cells	MDSCs and Tregs	Immunosuppression	The tumor stroma can attract immunosuppressive cell types like MDSCs and Tregs, which suppress cytotoxic T cells and NK cells.
Metabolic reprogramming	Tumor cells and stromal cells	Immunosuppression	Tumor cells and stromal cells can alter the metabolic landscape of the TME, creating conditions like hypoxia and nutrient deprivation that negatively impact immune cell function.
The modulation of extracellular matrix (ECM)	CAFs	Creates a physical barrier	CAFs can remodel the ECM, creating a physical barrier that hinders immune cell infiltration and access to tumor cells.
Cell polarization	TAMs, CD4+ T cells, MDSCs, and DCs	Immunosuppression	TAMs adopt an M2 polarization state, CD4+ T cells can be polarized into Tregs, MDSCs suppress T cell function, and DCs can become tolerogenic.

**Table 6 biomedicines-11-02411-t006:** LncRNAs involved in immune regulation and inflammation in cancer cell–tumor stroma interactions.

lncRNA Name (Genome Type)	Mechanism of Action	Effect on Cancer Progression	Interaction with Tumor Microenvironment
CCAT1 (Intergenic)	Influences inflammation, angiogenesis, and immune regulation via the microRNA-138-5p–HMGA1 axis in exosomes. Promotes immune cell polarization and pro-inflammatory cytokine release.	Promoting	Mediates angiogenesis and influences immune interactions in CRC.
CCAT2 (Sense)	Facilitates progression via PI3K/AKT/mTOR signaling. Enhances growth and metastasis via interactions with TAF15 to stimulate RAB14 transcription, triggering AKT/GSK3β signaling. Modulates the hsa-miR-145-5p/AKT3/mTOR axis in MCF7 cells.	Promoting	Plays a role in tumor–stroma immune interplay.
CRNDE (Intergenic)	Influences inflammation and immune evasion by releasing immunosuppressive factors, inducing T cell exhaustion, and recruiting regulatory immune cells. Activates NF-κB and JAK/STAT signaling. Participates in Th17 differentiation via CRNDE-h isoform interactions with RORγt.	Promoting	Mediates tumor–stroma immune interplay.
H19 (Intergenic)	Guides inflammation, pro-inflammatory cytokine release, and extracellular matrix remodeling via the upregulation of TNF-α. Involved in the STAT3 pathway.	Promoting	Facilitates interactions with immune cells in CRC.
HOTAIR (Antisense)	Connected with tumor grade and prognosis. Influences B cells toward a regulatory role via PDL1, suppressing CD8+ T cell activity. May stimulate pro-inflammatory cytokines and extracellular matrix remodeling.	Promoting	Suppresses CD8+ T cell activity.
HULC (Antisense)	Influences immune response, enhancing EZH2-H3K27me3 enrichment, and targets miR-613 and miR-488, promoting cell proliferation and suppressing p53 expression, which may facilitate tumor growth and metastasis.	Promoting	Influences immune response, promotes tumor growth and metastasis.
LINC00461 (Intergenic)	Mixed effects on CRC development and immunity. Promotes tumor growth and proliferation via the miR-323b-3p/NFIB axis. Acts as a competitive endogenous RNA (ceRNA) for PHLPP2, a colon cancer tumor suppressor.	Mixed	Influences cell migration, invasion, and transition, and the epithelial–mesenchymal transition.
lnc-ATB (Intergenic)	Involved in cancer progression, particularly in CRC, stimulating the release of pro-inflammatory cytokines and enhancing metastasis through pathways involving CDK2 and miR-200c.	Promoting	Enhances cancer metastasis and induces the EMT.
lnc-EGFR or EGILA (Antisense)	Facilitates immune evasion in CRC through the EGFR signaling pathway, potentially inducing T cell exhaustion and enhancing Treg differentiation.	Promoting	Enhances tumor immune escape.
MALAT1 (Intergenic)	Modulates T cell function, induces pro-inflammatory cytokines, and plays roles in immune evasion and inflammation in various cancers, potentially impacting NF-κB signaling.	Promoting	Suppresses innate and adaptive immune responses.
NEAT1 (Intergenic)	Contributes to immunosuppression and stemness maintenance through the regulation of ALDH1 and c-Myc. NEAT1 also influences metabolic and mitochondrial homeostasis, Fosters immune evasion through its expression in M2-polarized tumor-associated macrophages.	Promoting	Promotes immune evasion.
PCAT-1 (Antisense)	Enhances pro-inflammatory cytokine secretion and affects cellular processes like proliferation, invasion, and apoptosis by targeting miR-149-5p.	Promoting	Fosters an inflammatory tumor microenvironment.
SNHG1 (Intergenic)	Promotes pro-inflammatory cytokine secretion, cell proliferation, migration, and the EMT. It is linked with the Wnt/β-catenin pathway and molecules like MYC and SLC3A2, affecting immune responses and tumor progression.	Mixed	Influences immune response and induces the EMT.
SNHG14 (Sense)	Promotes CRC progression by negatively regulating EPHA7 through an EZH2-dependent pathway, enhancing methylation on the EPHA7 promoter and stabilizing EZH2 mRNA by interacting with FUS and freeing it from miR-186-5p-induced silence.	Promoting	Regulator of the immune response.
SOX2OT (Sense)	Related to inflammation and oncogenesis. It regulates pro-inflammatory cytokines and is linked with SOX2. Its silencing suppresses CRC cell growth and alters miR-194-5p	Mixed	Affects cell behavior in the TME.
TUG1 (Intergenic)	Interacts with the miR-138-5p/ZEB2 axis, promoting the EMT and, on the other hand, an immunosuppressive environment.	Mixed	Affects cancer cell behavior in the TME.
GAS5 (Antisense)	Downregulates IL-10 and VEGF-A via the NF-κB and Erk1/2 pathways. By inhibiting these cytokines, GAS5 suppresses tumor cell proliferation and promotes the tumor-suppressive function of M1 macrophages	Inhibiting	Reduces tumor immune escape, potentially reducing angiogenesis.

## Data Availability

All data and materials are available upon reasonable request. Address the request to N.K.S. (Nikolay K. Shakhpazyan) (email: nshakhpazyan@gmail.com) Avtsyn Research Institute of Human Morphology, Petrovsky National Research Center of Surgery, 119435 Moscow, Russia.

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
