# Peer review of "Long Non-Coding RNAs in Colorectal Cancer: Navigating the Intersections of Immunity, Intercellular Communication, and Therapeutic Potential"

_biomedicines, 2023, doi:10.3390/biomedicines11092411_

Round 1

Reviewer 1 Report

In this manuscript, the authors discuss the role of lncRNAs in the microenvironment of colorectal cancer, with particular emphasis on immunity and intercellular communication, and attempt to provide therapeutic guidance. The lncRNAs have been rapidly studied in recent years, and discussing them in the context of colorectal cancer will be significant from the perspective of understanding the disease pathogenesis and treatment. I commend you for summarizing, organizing, and discussing a wide range of literature.

However, the manuscript currently has several concerns that require improvement. I, therefore, request that the authors revise the manuscript by the comments.

Major comments

1. In many cases, only the consequences of lncRNAs are discussed and there is little explanation of how lncRNAs work in the way they do. The description of what molecules the lncRNAs act on and how they act should also be included.

2. In line114, the authors described that the CMS2 subtype associated with lncRNAs such as CCAT1. Are there commonalities between the lncRNAs associated with CMS2? If so, what are they like? Also, are there commonalities between the associated lncRNAs for other CMS subtypes?

1.     There is little explanation of the origin and mechanism of action of each lncRNA. At least for the lncRNAs listed in Tables 2 and 4, a table should be added explaining which gene they are encoded by, what type of lncRNA they are (intronic, exonic, etc.), and which molecules they target by what mechanism of action.

Minor comments

1.     For the CMS of CRCs, the authors should cite papers from international consortia that have been identified. Justin Guinney, et al., Nat Med. 2015 Nov; 21(11): 1350–1356. doi 10.1038/nm.3967

2.     Although this manuscript is focused on colorectal cancer, there are scattered citations on MCF7 and other cancer types. Although sources are natural as there are common phenomena in cancer, such as signal pathways, the authors should present careful descriptions to avoid misunderstandings.

3.     In line 82, the authors described mRNA splicing, stability, and translation, with MALAT1 being an example [15]. Ref. 15 seems to mention the control of alternative splicing, but is MALAT1 also involved in the control of stability and translation?

Author Response

Major comments

1. In many cases, only the consequences of lncRNAs are discussed and there is little explanation of how lncRNAs work in the way they do. The description of what molecules the lncRNAs act on and how they act should also be included.

Thank you for the substantial comment. Changes have been made that supplement and reveal the molecular mechanisms of the described lncRNAs.

2. In line114, the authors described that the CMS2 subtype associated with lncRNAs such as CCAT1. Are there commonalities between the lncRNAs associated with CMS2? If so, what are they like? Also, are there commonalities between the associated lncRNAs for other CMS subtypes?

We have made changes in Chapter 3, showing the similarity of lncRNAs where it manifests within the CMS groups.

3. There is little explanation of the origin and mechanism of action of each lncRNA. At least for the lncRNAs listed in Tables 2 and 4, a table should be added explaining which gene they are encoded by, what type of lncRNA they are (intronic, exonic, etc.), and which molecules they target by what mechanism of action.

On your recommendation, we have added two tables - Table 1 contains the genomic classification of lncRNAs, Table 2 - Overview of Long Non-Coding RNA (lncRNA) Functions and Mechanisms in Cellular Processes. These tables are made to facilitate the description of lncRNAs later in the manuscript. The changes you recommended have been incorporated into the tables (now Tables 4 and 6).

Minor comments

  1. For the CMS of CRCs, the authors should cite papers from international consortia that have been identified. Justin Guinney, et al., Nat Med. 2015 Nov; 21(11): 1350–1356. doi 10.1038/nm.3967

The literary source has been added, and it appears as number 49 in the reference list and in the text.

  1. Although this manuscript is focused on colorectal cancer, there are scattered citations on MCF7 and other cancer types. Although sources are natural as there are common phenomena in cancer, such as signal pathways, the authors should present careful descriptions to avoid misunderstandings.

The corresponding explanation has been added to the text -line 341.

3. References to sources confirming the involvement of MALAT1 in RNA stability and translation have been added - line 100.

Reviewer 2 Report

1. Introduction should be modified, in present form is divided into separate but not connected paragraphs.

2.  The authors decided to focus on two mechanisms inflammation and intracelullar communication, but construction of these paragrpahs is not similar. The paragraph about immunity and  inflammation is unproportionally  long. The next section (6) is only one page. It should be changed according to the same template.

3. The Authors should add information from bioinformatical analysis, as the evidences that selected and described in th manuscript molecules are important.

4. The last section about "Exploring Strategies for Manipulating lncRNAs to Enhance Anti-Tumor Immunity 531 in CRC Patients" also should be modified. Different strategies should be bolded in the manuscript, additional figures could be added.

The Authors should add paragraph about clinical studies as a summary of this part of manuscript.

-

Author Response

  1. Introduction should be modified, in present form is divided into separate but not connected paragraphs.

Based on your recommendation, the Introduction chapter has been altered to improve the coherence of the text.

2.  The authors decided to focus on two mechanisms inflammation and intracelullar communication, but construction of these paragrpahs is not similar. The paragraph about immunity and  inflammation is unproportionally  long. The next section (6) is only one page. It should be changed according to the same template.

Chapter 6, 'Unraveling the Complexity: Interplay of lncRNAs and Other ncRNAs in Cell-to-Cell Communication within the CRC Microenvironment,' has been supplemented per your recommendation.

3. The Authors should add information from bioinformatical analysis, as the evidences that selected and described in th manuscript molecules are important.

We have added information about bioinformatic approaches that are used to investigate issues related to the functions of lncRNAs in Chapter 2, 'Fundamentals of Long Non-coding RNAs,' and Chapter 6 (line 563).

4. The last section about "Exploring Strategies for Manipulating lncRNAs to Enhance Anti-Tumor Immunity 531 in CRC Patients" also should be modified. Different strategies should be bolded in the manuscript, additional figures could be added.

The Authors should add paragraph about clinical studies as a summary of this part of manuscript.

The chapter has been modified, divided into subsections, and illustrations have been added. Paragraphs about clinical studies have been included. Unfortunately, there is little information on clinical studies concerning lncRNAs in patients with colorectal cancer; all the studies we found play a diagnostic role.

Round 2

Reviewer 1 Report

The revised manuscript has been improved by addressing the comments. Although some of the tables are not easy to understand, I conclude that, in general, the level of detail is acceptable for publication. The authors are to be congratulated for their efforts. I, therefore, consider that this revised manuscript should be accepted.

Reviewer 2 Report

The manusrcipt has been revised and can be published